# In Silico Identification and Characterization of Satellite DNAs in 23 *Drosophila* Species from the *Montium* Group

**DOI:** 10.3390/genes14020300

**Published:** 2023-01-23

**Authors:** Bráulio S. M. L. Silva, Agnello C. R. Picorelli, Gustavo C. S. Kuhn

**Affiliations:** Department of Genetics, Ecology and Evolution, Federal University of Minas Gerais, Belo Horizonte 31270-901, Brazil

**Keywords:** satellite DNA, tandem repeats, repetitive DNA, Helitrons, genome evolution, TAREAN, *Drosophila*, *montium* group

## Abstract

Satellite DNA (satDNA) is a class of tandemly repeated non-protein coding DNA sequences which can be found in abundance in eukaryotic genomes. They can be functional, impact the genomic architecture in many ways, and their rapid evolution has consequences for species diversification. We took advantage of the recent availability of sequenced genomes from 23 *Drosophila* species from the *montium* group to study their satDNA landscape. For this purpose, we used publicly available whole-genome sequencing Illumina reads and the TAREAN (tandem repeat analyzer) pipeline. We provide the characterization of 101 non-homologous satDNA families in this group, 93 of which are described here for the first time. Their repeat units vary in size from 4 bp to 1897 bp, but most satDNAs show repeat units < 100 bp long and, among them, repeats ≤ 10 bp are the most frequent ones. The genomic contribution of the satDNAs ranges from ~1.4% to 21.6%. There is no significant correlation between satDNA content and genome sizes in the 23 species. We also found that at least one satDNA originated from an expansion of the central tandem repeats (CTRs) present inside a Helitron transposon. Finally, some satDNAs may be useful as taxonomic markers for the identification of species or subgroups within the group.

## 1. Introduction

Eukaryotic genomes are enriched by a great number and variety of non-protein-coding repetitive DNA elements. The genomic fraction made by these elements varies between species, but it can frequently reach >50% in several animal and plant species. They can be found dispersed along the genome, in forms such as transposable elements (TEs), and/or in tandem organization, as microsatellites, minisatellites and satellite DNAs (satDNAs) [1,2].

Individual satDNA families typically reach more than 10^3^ copies in the genome. These copies form large, in some cases Mb-size arrays, that are mainly concentrated in heterochromatin-rich regions of the chromosomes, such as the (peri)centromeric and subtelomeric regions [2,3,4,5,6]. However, occasionally they may also be present along the euchromatin in the form of small arrays (with 1–20 tandem repeats) [7,8,9]. In contrast, microsatellites and minisatellites are less repetitive (<10^3^ copies), and their shorter arrays are in a scattered distribution throughout the genome. Concerning repeat length (i.e., monomer size), microsatellites are usually in the range of few base pairs to <10 bp, minisatellites between 10 and 200 bp, and satellites between 2 bp to > several hundred bp [3,4].

Once considered fully “junk DNA” in the past, it is now recognized that satDNAs (or a fraction of them) may participate in important genomic functions, such as gene regulation and chromatin modulation [10,11], spatial chromosome organization [12,13,14], and centromeric architecture [15]. Furthermore, satDNAs contribute to the generation of genome size differences among species and may also be related to the origin of chromosome rearrangements [16,17]. SatDNAs evolve rapidly and may also contribute to the establishment of genetic incompatibilities and reproductive isolation between incipient species [18]. Therefore, there is no doubt today that the study of satDNAs is highly relevant in the context of functional and evolutionary genomics [6,16,19,20]. 

Species from the genus *Drosophila* have been extensively used as model to address several aspects related to satDNA structure, organization, function, evolution, and impact on speciation (e.g., [11,14,18,21,22,23,24,25,26]). In the last 10 years, these studies have been fostered by the large number of *Drosophila* species with sequenced genomes available, and the concomitant development of several new bioinformatic tools specifically designed for the identification of satDNAs, such as the TAREAN (Tandem Repeat Analyzer) pipeline [27]. More recently, the genomes of 23 *Drosophila* species from the *montium* group have been sequenced, but no information about their satDNAs has been reported to date [28]. 

The *montium* group, with 71 Asian and Australasian species and 23 African species, is the largest clade within the subgenus *Sophophora*. Based on the analyses of morphological (male abdominal pigmentation and genitalia) and chorological traits, the group can be subdivided into seven subgroups (*parvula*, *montium*, *punjabiensis*, *serrata*, *kikkawai*, *seguyi,* and *orosa*) whose phylogenetic relationships have been inferred from three nuclear genes and one mitochondrial gene [29] (Figure 1A). 

A recent phylogenetic analysis, performed using 60 genes made by Conner et al. [30], confirmed the monophyly of the seven *montium* subgroups proposed by Yassin [29]. However, this later study showed that the *montium* subgroup is the most basal subgroup in the phylogeny. Moreover, it showed that the *punjabiensis* subgroup is closer to the *seguyi* subgroup and that the *kikkawai* subgroup is the third most basal clade of the group (Figure 1B) [30].

The basic metaphase karyotype of species from the *montium* group consists of one pair of sex chromosomes, two pairs of acrocentric chromosomes and one pair of microchromosomes [32]. The only reported changes in the metaphase karyotype configuration among species concern the variation in the amount of heterochromatin present in the microchromosomes and/or in the Y chromosome and, to a lesser extent, in the X chromosome [32,33]. As found in *Drosophila* and other eukaryotic species, changes in the amount of heterochromatin may be directly connected to expansions or contractions of satDNAs, which is the most abundant component of heterochromatin [34,35,36,37,38].

In the present work, we aimed to characterize the satDNA landscape of 23 species from the *montium* group. We first used the TAREAN pipeline to identify and quantify putative satDNAs in the 23 species, and then created a more conservative “satDNA filter” to select only the families sharing more attributes with satDNAs. We ended up with 142 satDNA clusters representing 101 non-homologous satDNA families. The data are discussed in terms of satDNA’s general structural features, its relationship to genome sizes, and its relationship to transposable elements. We expect that our collection of identified satDNAs will be useful for future studies concerning genome annotation and genome/chromosome evolution in the *montium* group. Additionally, some satDNA families may be useful as potential taxonomic markers for the identification of species or specific clades/subgroups within the *montium* species group. 

## 2. Materials and Methods

### Satellite DNA Identification

TAREAN is a computational pipeline used for the unsupervised identification of satDNAs from unassembled short-read sequences [27]. In this study, we used publicly available Illumina paired-end sequencing raw data from 23 species (females) from the *montium* group on NCBI (Accession: PRJNA554346—ID: 554346) [28] (Table 1). TAREAN analyses were performed on the Galaxy Platform [39]. We first measured the reads quality with the “FASTQC” tool and converted all the sequences to a single fastqsanger format with the “FASTQ Groomer” tool (Sanger and Illumina 1.8+). After the removal of adapters and reads presenting more than 5% of low-quality bases (Phred cutoff < 10), the reads were trimmed to 100 bp along with the “Preprocessing of fastq paired-reads” tool. The resulting file, with interlaced filtered paired-end reads, was used as an input for the TAREAN pipeline, with the following settings: “read sampling: no—advanced options: yes—perform cluster merging: yes—use custom repeat database: no—cluster size threshold for detailed analysis: 0.01—perform automatic filtering of abundant satellite repeats: no—keep original read names: no—similarity search options: masking of low complexity repeats disabled—select queue: basic”. The resulting archives, containing the putative satDNA clusters, were downloaded for a more detailed investigation. Only putative satDNA clusters, showing a minimum of 0.1% genomic contribution to at least one species of the genomic DNA, were selected for further analysis. Considering typical genome sizes of species from the *montium* group as being around 196.3 Mb, 0.1% genomic contribution corresponds to ~1,963,000 copies of satDNA with 10 bp or ~196,300 copies of a satDNA with 100 bp repeats.

The estimated genomic proportion of each putative satDNA cluster is initially presented in the TAREAN results as the proportion of the reads in each cluster concerning the number of all analyzed reads. However, the analyzed reads by TAREAN may contain organellar DNA and contaminant DNA. For this reason, we checked all clusters retrieved by TAREAN in each species and removed (when present) the reads from clusters corresponding to mitochondrial DNA and contaminants. Next, we recalculated the genomic proportion of each putative satDNA based on the number of total reads representing only nuclear sequences, as proposed by Novák et al. [40].

The TAREAN pipeline classifies the clusters with putative satDNA sequences into two categories: satellites with high confidence (HC) and satellites with low confidence (LC). These categories are determined according to the “Connected component index (C)”, which indicates clusters formed by tandem repeat sequences, and “Pair completeness index (P)”, which measures the length of continuous tandem arrays [27]. Another important aspect of TAREAN is that the pipeline groups the reads into clusters according to their sequence similarity. Similar reads form graphs represented by nodes and connecting edges, and graphs presenting globular shapes are likely constituted by satDNAs [27]. 

After selecting the satDNA clusters with more than 0.1% genomic contribution, we developed a second satDNA “filter” in which the selected clusters should comply with three out of the four following parameters: *c* value > 0.9, *p* value > 0.8, high confidence and circular graph layout (e.g., Appendix A). After this cut-off analysis, we conducted further analyses on the remaining satDNA clusters and their corresponding consensus sequences provided in the TAREAN results.

For the identification of homologous satDNAs shared by two or more species, we created a custom database on the Geneious Software [41] containing all consensus sequences from the selected satDNAs. We then used each satDNA consensus for MEGABLAST searches of the custom database (maximum e-value = 1 × 10^−5^; gap cost = linear; threshold = 0%; majority: most common bases, fewest ambiguities). Figure 2 shows the workflow chart of our study.

## 3. Results and Discussion

### 3.1. Identification of Satellite DNA Families in the Montium Group

The TAREAN analysis first retrieved 397 clusters, identified as putative satDNAs in the 23 species from the *montium* group, namely, 245 with high confidence (HC) and 152 with low confidence (LC) (Table 1). After filtering the clusters using our custom satDNA filter, the number of satDNAs narrowed down to 142 (Figure 2). Then, we created a custom database containing consensus sequences of each one of these 142 satDNAs and conducted MEGABLAST searches using each consensus sequence against our whole custom database. We found that the 142 satDNAs correspond to 101 satDNA non-homologous families, which have been numbered dmgsat-1 to dmgsat-101 (Appendix A). The consensus sequences of these satDNAs can be found in Appendix A.

It is assumed that the *c* and *p* values retrieved from TAREAN analyses are important parameters for a reliable satDNA identification, as both values together give a good indication that the identified clusters correspond to repeats, organized as long and continuous satDNA-like arrays. For example, several studies showed that clusters with high genome proportion (>1%), high *c* and *p* values (>0.98) and high satellite probability (>0.95) correspond to typical satDNAs sequences that are located on the centromeric and/or pericentromeric regions of the chromosomes [42,43,44,45]. Accordingly, all satDNAs selected for our study have *c* and *p* values near or above 0.9 in at least one species (Appendix A).

To our knowledge, from all the 101 satDNA families we found in the *montium* group, only 8 families showed any homology with previously described satDNAs in other *Drosophila* species (Appendix A): the dmgsat-14 and dmgsat-67 satDNA families share sequence homology to the “1.688” satellite DNA [7,9,46], and the dmgsat-52 satDNA family is homologous to the “1.669” satDNA [22,47,48]. Recently, de Lima and Ruiz-Ruano [49] reported an in silico characterization of satellite DNAs in two species from the *montium* group, *D. burlai* and *D. leontia*, using the RepeatExplorer pipeline. We have noted that five satDNAs reported here are homologous with satDNAs reported in de Lima and Ruiz-Ruano [49]: the dmgsat-10 and dmgsat-11 are homologous to “DleoSat1-41“ and “Dleosat4-109“ from *D. leontia*, respectively, and dmgsat-51, dmgsat-61, and dmgsat-85 are homologous to “DburSat3-9“, “DburSat2-300“ and “DburSat1-135“ from *D. burlai*, respectively.

### 3.2. Satellite DNAs in the Montium Group: General Structural Features

There is an extensive variation in repeat length in the 101 satDNAs found in species of the *montium* group, from only 4 bp (dmgsat-35 from *D. triauraria*) to 1897 bp (dmgsat-63 from *D. burlai*) (Figure 3). However, most satDNAs (89%) are within the range of the most common repeat length found in *Drosophila* (from <10 bp to 400 bp) [26,50,51]. 

Most satDNAs (60%) showed repeats shorter than 100 bp (Figure 3A). To better assess the repeat length variation of the 65 satDNAs with repeats shorter than 100 bp, we further subdivided this class into 10 intervals of 10 bp each (Figure 3B). Most short satDNA families have repeat sizes shorter than 10 bp (52.3%) (Figure 3B).

Therefore, we concluded that the 23 species genomes from the *montium* group investigated in our study are enriched with satDNAs consisting of short (<100 bp) tandem repeats, especially in the range of 1–9 bp. The presence of satDNAs with short repeat sizes in *Drosophila* is not rare. For example, abundant satDNA families with repeats 7 bp long are found in *Drosophila virilis* [52,53], and *D. melanogaster* has several satDNAs with repeats in the range between 5 bp to 10 bp [22].

We found that 78 satDNA families have an AT content > 60% (Figure 4). This number represents 76.5% of the total number of families. Therefore, our findings show that satDNA sequences, present in species from the *montium* group, are also mostly AT rich, as found previously for other groups and species of *Drosophila* [22,48,49,52,53,54,55].

### 3.3. Satellite DNA Abundance and Relationship with Genome Sizes

SatDNAs usually account for more than 20% of the genomic DNA in species from the *Drosophila* genus [56], as in *D. melanogaster*, and up to 70% in some Hawaiian *Drosophila* [57], but less than 3% in species from the *repleta* group [26]. In *Drosophila* and many organisms, there is a positive correlation between satDNA content and genome sizes [49,56]. 

The genome sizes in the 23 *Drosophila* studied species from the *montium* group were estimated by Bronski et al. [28] and they range from 155.1 Mb (*D. bocki*) to 223.4 Mb (*D. mayri*). Based on the TAREAN results, our estimated satDNA contribution to total genomic DNA ranges from 1.40% (*D. watanabei*) to 21.65% (*D. pectinifera*) (Figure 5). Such 16-fold variation does not match the 1.4-fold variation found among genome sizes. Accordingly, we found no significant positive correlation between satDNA abundance and genome sizes (Figure 6). We also performed correlation tests between genome sizes and all initial 397 putative satDNA clusters returned by the TAREAN analysis (Figure 2), but again we found no significant correlations (Appendix A). 

Considering that Bronski et al. [28] found a strong positive correlation between genome sizes and the whole repetitive DNA content across all the 23 *montium* genomes, we suggest that other repetitive DNAs, probably transposable elements, are the main repetitive DNAs promoting genome size variation in this group of species. In accordance with this hypothesis, a recent study revealed that TE abundance, but not satDNAs, is positively correlated with genome sizes in *Drosophila* species from the *Sophophora* genus, where species from the *montium* group also belong [49].

### 3.4. Satellite DNA Distribution across the Montium Phylogeny

Studies in several species of eukaryotes have revealed that satDNAs are among the fastest evolving components of the genome. This assumption is supported by the large number of satDNAs that are found restricted to a few closely related species, or even to a single species [23,26,58]. 

None of the 101 satDNA families we described here are present in all 23 analyzed species from the *montium* group. This result is not surprising, considering that the common ancestor of the *montium* group lived in Asia more than 19 Mya [29]. In fact, our results showed that most satDNAs families (83%) seem to be restricted to a single species. However, our results obtained with TAREAN do not exclude the possibility that homologous low-copy number, or highly variable repeats, are present in additional species.

From our collection of 101 satDNAs, only 17 are shared by at least two species. The distribution of these 17 satDNAs across the montium group phylogeny is mostly in accordance with the phylogenies proposed by Yassin [29] and Conner et al. [30] at the subgroup level (Figure 7). Several satDNAs are also restricted to species from the same subgroup, such as dmgsat-1, dmgsat-2, and dmgsat-3 from the *montium* subgroup, dmgsatDNA-4 in the *punjabiensis* subgroup, dmgsat-5, dmgsat-6, dmgsat-7, dmgsat-8, dmgsat-9 in the *serrata* subgroup, dmgsat-10, dmgsat-11, and dmgsat-12 in the *kikkawai* subgroup, and dmgsat-14 in species from the *seguyi* subgroup. The remaining satDNAs (dmgsat-13, dmgsat-15, dmgsat-16, and dmgsat-17) are shared between species from different subgroups.

### 3.5. Satellite DNA Emergence from DINEs

The *Drosophila* interspersed elements, or DINEs, are abundant (>1000 copies) transposable elements (TEs) found in several *Drosophila* species [59]. They are classified as nonautonomous variants of Helitrons, called Helentrons, and their general structure consist of two conserved blocks (A and B) separated by central tandem repeats (CTRs) (Figure 8A) [60].

Dias et al. [62] identified a DINE variant, named DINE-TR1, to be present in several *Drosophila* species and even outside the genus (in *Bactrocera tryoni*). This DINE-TR1 has CTRs of ~150 bp which are homologous across species. Interestingly, these CTRs have undergone amplification to satDNA-like arrays independently twice across the *Drosophila* phylogeny, both in the ancestral species of *D. virilis* and *D. americana*, and also in *D. biarmipes* [62].

In the present work, we investigated if our collection of 101 satDNA families from the *montium* group shares homology with transposable elements, specially Helitrons. For this purpose, we used our whole collection of satDNA consensus sequences from each satDNA family to screen the CENSOR database on Repbase [63] for homologous known TEs. The results are shown in Appendix A. We found that 12 satDNA families (dmgsat-1, dmgsat-7, dmgsat-8, dmgsat-14, dmgsat-20, dmgsat-22, dmgsat-41, dmgsat-67, dmgsat-79, dmgsat-81, dmgsat-84, and dmgsat-91) share regions of DNA sequence identity > 70% to *Drosophila* Helitrons (Appendix A). From these satDNAs, we selected dmgsat-7, present in *D. mayri* and *D. serrata* from the *serrata* subgroup, for further in-deep analysis. This was done because its repeat units are very similar in length (~150 bp) to the CTRs present in DINE-TR1. We found that dmgsat-7 consensus sequences are homologous to CTRs present in DINE-TR1 from *D. biarmipes* and *D. virilis,* suggesting that dmgsat-7 is another case of satDNA emergence from DINE-TR1-expanded CTRs (Figure 8B). Interestingly, high sequence identity is limited to the first 30 bp, which possibly indicates the participation of this conserved segment in some functional role, as proposed by Dias et al. [62]. The dmgsat-7 genomic proportion is high in *D. mayri* (2.5%) and *D. serrata* (2.0%) (Figure 7 and Appendix A). These values are close to the genomic proportion of the expanded DINE-TR1 CTRs found in *D. virilis* (1.6%) and *D. americana* (2.2%) [43]. To date, *Drosophila serrata* is the only species from the *serrata* subgroup whose genome has been sequenced with long-read sequencing technology (GenBank: GCA_002093755.1) [64], which allowed us to investigate the size of dmgsat-7 arrays in more detail. Accordingly, we were able to detect dmgsat-7 uninterrupted arrays up to ~ 82.6 kb (~ 540 tandem copies) in *D. serrata* (Appendix A). 

In summary, our results show that dmgsat-7 is another example of a satDNA derived from the expansion of internal tandem repeats present in DINE-TR1, reinforcing the importance of DINE-TR1 as a potential source for the emergence of satDNAs, as previously suggested [62].

## 4. Conclusions

With the advent of a new generation DNA sequencing techniques, new bioinformatics tools have been providing efficient ways to identify and classify repetitive DNAs [65,66]. In this context, the TAREAN pipeline was designed as a tool for the identification of satDNA sequences from unassembled short reads. Several studies show that TAREAN is an efficient method for the identification of satDNAs from eukaryotic genomes (e.g., [27,43,45,67,68]).

TAREAN analyses, combined with subsequent manual curation, revealed the presence of 101 satDNAs in 23 *Drosophila* species from the *montium* group, most of them being reported here for the first time. The data presented are expected to provide the framework for future genomic/satellite DNA studies in this group. In particular, the only reported changes in the karyotype configuration of species from the *montium* group concern changes in the amount of heterochromatin [32,33]. It will be interesting to investigate whether these changes are associated with the satDNAs described here.

## Figures and Tables

**Figure 1 genes-14-00300-f001:**
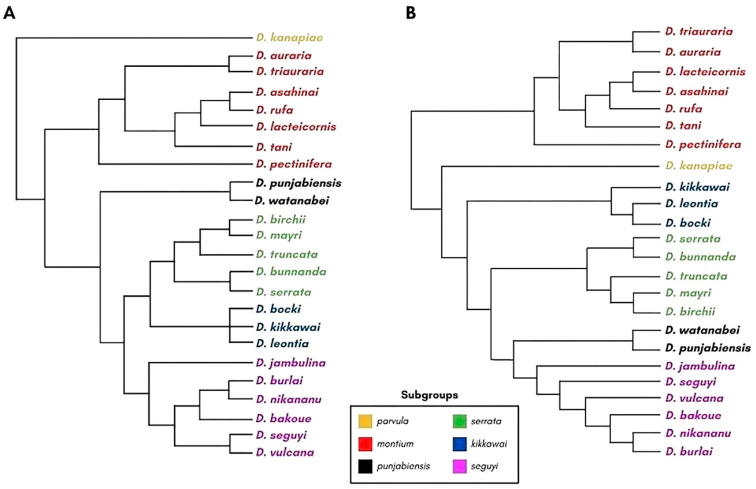
Phylogenetic relationships among subgroups from the *montium* group. Only the species investigated in the present work are shown. (**A**) Phylogeny based on three nuclear genes and one mitochondrial gene (adapted from Yassin [29]). (**B**) Phylogeny based on 60 genes (adapted from Conner et al. [30]). The branch lengths do not correspond to evolutionary distances. The phylogenetic trees were reconstructed using the Archaeopteryx software [31].

**Figure 2 genes-14-00300-f002:**
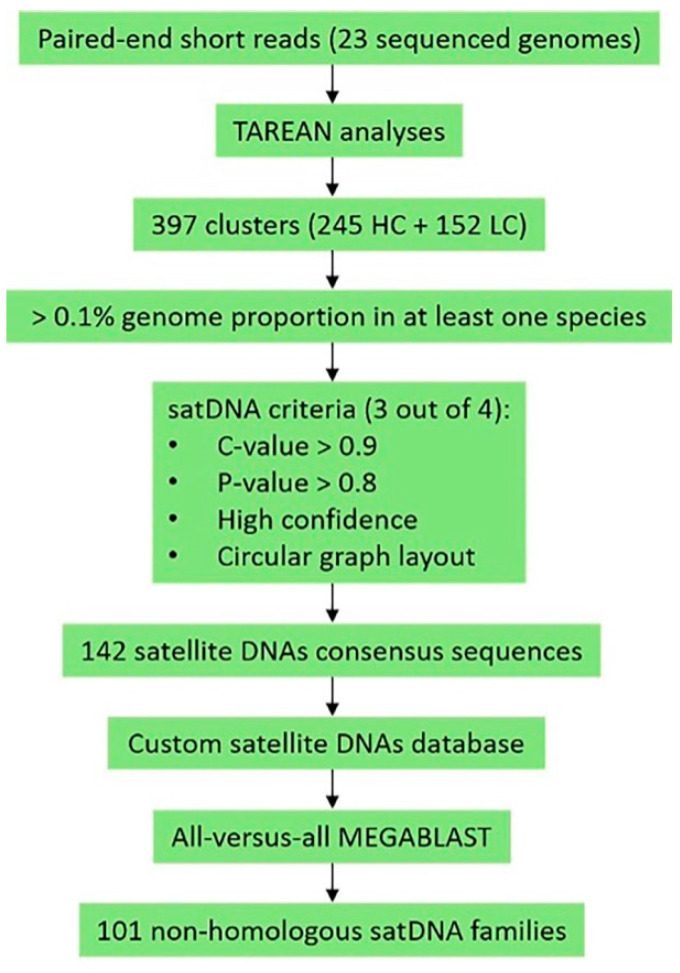
Workflow used for satellite DNA (satDNA) identification in the 23 sequenced *montium* genomes.

**Figure 3 genes-14-00300-f003:**
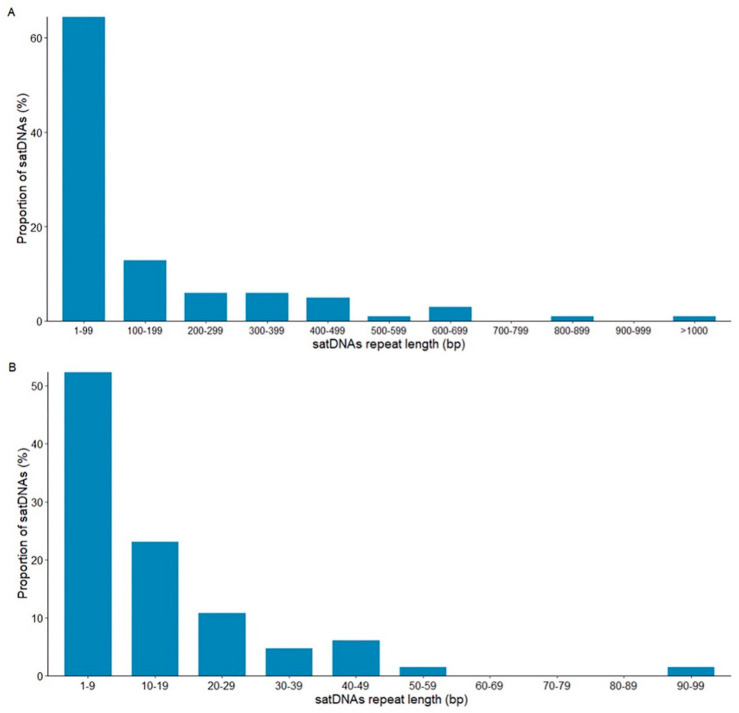
SatDNAs repeat length (monomer size) in the *montium* group. (**A**) The repeat length of all 101 satDNA families identified in the present work. (**B**) Repeat length of the 65 satDNA families featuring less than 100 bp long repeats.

**Figure 4 genes-14-00300-f004:**
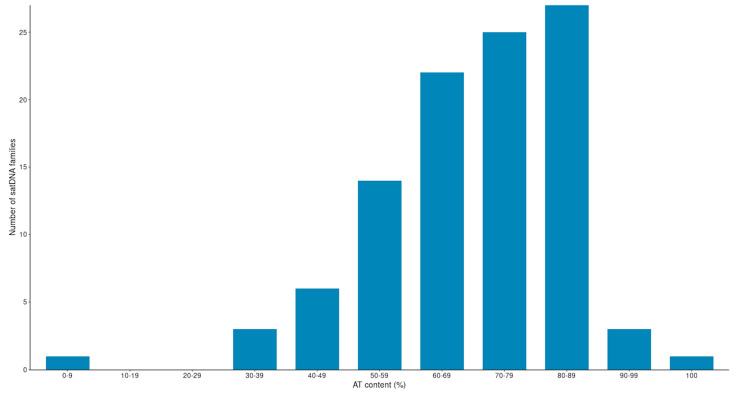
AT content of the 101 satDNA families identified in the *montium* group.

**Figure 5 genes-14-00300-f005:**
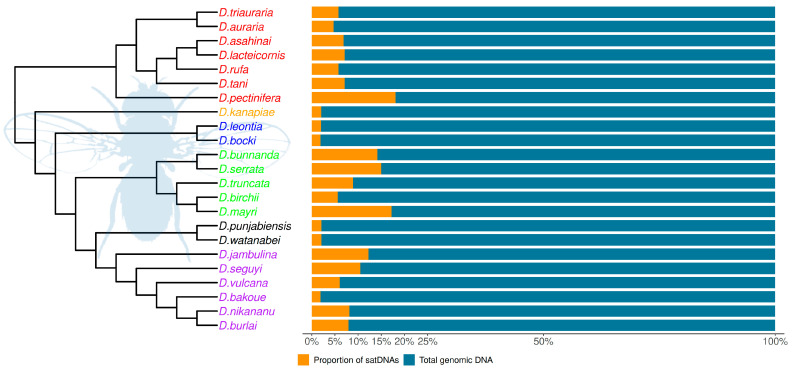
SatDNA genomic proportions in the 23 analyzed species from the *montium* group. The phylogenetic tree was reconstructed according to Conner et al. [30]. Species names are colored according to the subgroups they belong to (see Figure 1).

**Figure 6 genes-14-00300-f006:**
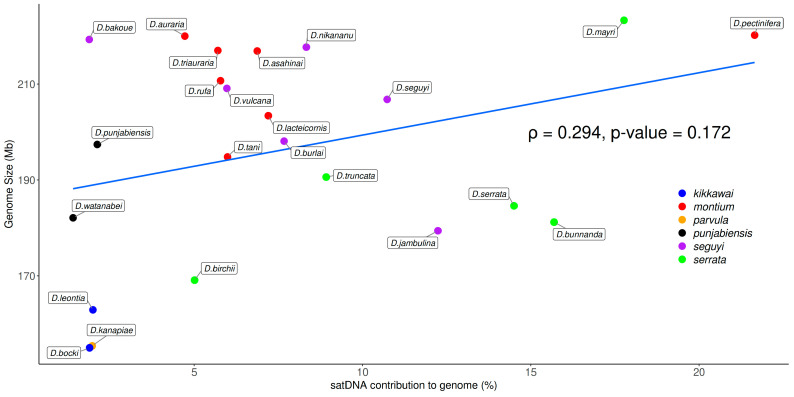
Correlation test between genome size and satDNAs contribution to genome in species from the *montium* group. The *p* value was obtained with Spearman’s correlation test.

**Figure 7 genes-14-00300-f007:**
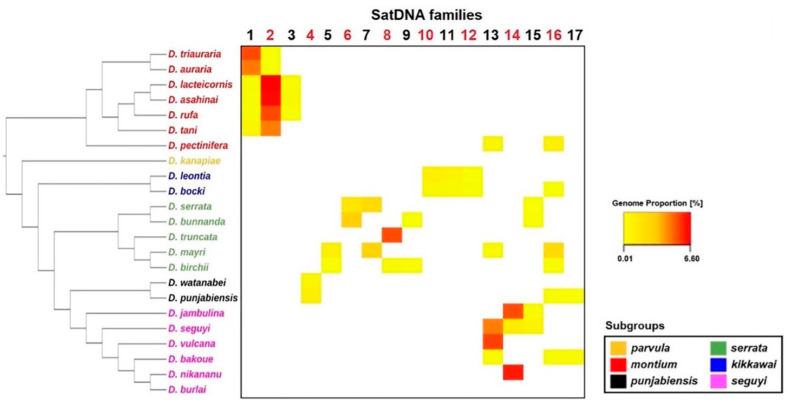
Heatmap showing the genomic proportion for each satDNA family shared by at least two species. The phylogenetic tree was reconstructed according to Conner et al. [30] using the Archaeopteryx software [31]. Species names are colored according to the subgroups they belong to (see Figure 1). The genomic proportion values for each satDNA are described in Appendix A.

**Figure 8 genes-14-00300-f008:**
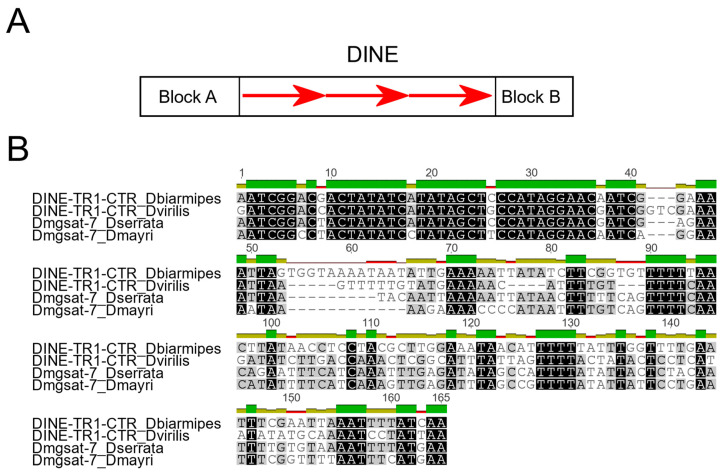
(**A**) General structure of DINEs, including DINE-TR1. Red arrows correspond to the central tandem repeats (CTRs). (**B**) Nucleotide sequence alignment (MUSCLE method) [61] containing DINE-TR1 CTR consensus sequences from D. *biarmipes* and *D. virilis* and the dmgsat-7 consensus sequences from *D. serrata* and *D. mayri*.

**Table 1 genes-14-00300-t001:** SatDNA-like clusters identified in the *montium* group by TAREAN, before and after filtering. HC = High confidence; LC = Low confidence.

Species	Subgroup	HC satDNAs (Before Filtering)	LC satDNAs (Before Filtering)	Final Number of satDNA-like Families (After Filtering)
*D. kanapiae*	*parvula*	13	9	4
*D. auraria*	*montium*	3	10	2
*D. triauraria*	*montium*	6	5	3
*D. asahinai*	*montium*	7	8	3
*D. rufa*	*montium*	4	7	3
*D. lacteicornis*	*montium*	6	5	3
*D. tani*	*montium*	7	9	4
*D. pectinifera*	*montium*	14	5	10
*D. punjabiensis*	*punjabiensis*	14	5	6
*D. watanabei*	*punjabiensis*	7	7	3
*D. birchii*	*serrata*	15	5	8
*D. mayri*	*serrata*	15	8	13
*D. truncata*	*serrata*	11	6	5
*D. bunnanda*	*serrata*	24	6	14
*D. serrata*	*serrata*	12	9	6
*D. bocki*	*kikkawai*	10	4	7
*D. leontia*	*kikkawai*	5	7	4
*D. jambulina*	*seguyi*	8	2	6
*D. burlai*	*seguyi*	11	7	7
*D. nikananu*	*seguyi*	6	6	3
*D. bakoue*	*seguyi*	25	8	9
*D. seguyi*	*seguyi*	17	6	12
*D. vulcana*	*seguyi*	5	8	3
Total		245	152	

## Data Availability

Not applicable.

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
