# Peer review of "In Silico Identification and Characterization of Satellite DNAs in 23 Drosophila Species from the Montium Group"

_genes, 2023, doi:10.3390/genes14020300_

Round 1
Reviewer 1 Report
The presented paper by Silva, Picorelli and Kuhn dealt with the determination of satellite DNA families within a species group of Drosophila.
The article is well presented, leading the reader easily through the objectives, results and conclusions. They offered some interesting points about the relationship between C-value and satDNA content in these Drosophila species, and the importance of satDNA variation on karyotypic evolution. It called my attention, their results on the liaisons between satDNA and Helitrons. Very interesting results and perfectly discussed.
I must point out some minor recommendations that I would suggest to the authors:
-Include in the figure one caption, more details about the presented trees (Fig1). The data is in the text, but in my opinion the figures must be self-explanatory.
-Line 80: "and, and"
- I recommend to authors that exclude from the analysis the clusters corresponding to mitochondrial DNA or other possible contamination. Since this affects the estimations of the TAREAN clusters. In case that authors actually did this, I encourage them to explain it in the methods section.
-The sentence within lines 194 and 196 must be rewritten. It is not clear. The idea is there but the sentence is not correctly stated.
-I found very strange to include the citation as part of the text. Such as in line 215. I would include "Bronski et al" and then the "[28]". Same thing is observed in line 285. But is even more strange to initiate the sentence with the citation as in line 266.
-Just as an idea: I understand that other repetitive sequences are not the main objetive of this paper. But the authors have the data (in the results of TAREAN or RepeatExplorer, also in galaxy), to discus about the relationship of genome content (C-value) and the total repetitive fraction of the genome (lines 215 and 218).
-Please join last three paragraphs. Since they are related to the same topic.
Overall, I would like to congratulate authors on their nicely presented and very interesting paper.
Author Response
We thank the reviewer for all the suggestions. Below we provide a point-by-point response to all comments:
- Include in the figure one caption, more details about the presented trees (Fig1). The data is in the text, but in my opinion the figures must be self-explanatory.
We have added more details on the legend of Figure 1, as requested.
- Line 80: "and, and"
Corrected in the new version.
- I recommend to authors that exclude from the analysis the clusters corresponding to mitochondrial DNA or other possible contamination. Since this affects the estimations of the TAREAN clusters. In case that authors actually did this, I encourage them to explain it in the methods section.
Following the suggestion of the reviewer, we made the exclusion of clusters corresponding to mitochondrial and contaminants and recalculated the new genome proportions for the remaining genomic clusters of putative satellite DNAs. We explained this step in the lines 131-138. The estimation of satDNA genomic contribution slightly changed with this new analysis. With the new data, we reconstructed Fig 5 and remade the correlation tests in figure 6. The main conclusions did not change. These new genomic proportion estimates for each satDNA have also been corrected on Table S1.
-The sentence within lines 194 and 196 must be rewritten. It is not clear. The idea is there but the sentence is not correctly stated.
This sentence was rewritten as follows (lines 254-256): “SatDNAs usually account for more than 20% of the genomic DNA in species from the Drosophila genus [55], as in D. melanogaster, and up to 70% in some Hawaiian Drosophila [56], but less than 3% in species from the repleta group [26]”
- I found very strange to include the citation as part of the text. Such as in line 215. I would include "Bronski et al" and then the "[28]". Same thing is observed in line 285. But is even more strange to initiate the sentence with the citation as in line 266.
We have corrected this issue in the revised version.
- Just as an idea: I understand that other repetitive sequences are not the main objetive of this paper. But the authors have the data (in the results of TAREAN or RepeatExplorer, also in galaxy), to discus about the relationship of genome content (C-value) and the total repetitive fraction of the genome (lines 215 and 218).
We agree on the importance of the discussion about the C-value and total fraction of repetitive DNAs. However, as pointed by the reviewer, this was not the focus of our work. Moreover, such analysis has already been made by Bronski et al. (2020), as we mentioned in Line 279-282.
- Please join last three paragraphs. Since they are related to the same topic.
We partially agree with the reviewer concerning this suggestion. We joined the first two mentioned paragraphs (lines 390-412), but the third, starting as “In summary..” (Line 413) was left as a final separated paragraph of this topic.
Reviewer 2 Report
This paper describes a fairly straight-forward analysis of the abundances of 101 satellite DNA sequences of 23 D. montium group species using short-read libraries. The work makes use of TAREAN software, which identifies tandem repeats within Illumina sequence reads. The authors find a wide range of the proportion of the genomes that are comprised of satellite repeats, ranging from ~1.4% to 22.2%. No particular correlates are found to identify any possible evolutionary reason for this range. Furthermore, there was no significant correlation between satDNA content and genome size across the 23 species. Only rarely was there any relation between satellite repeats and transposable element sequences. While there was not a strong phylogenetic signal of the satellite repeats, the signature of satellite compositions was sufficiently distinctive that species could be identified to some extent by their satellite abundances.
The analysis is reasonably well motivated and clearly described, and while the authors do not extend the methodology, they apply it well and the results appear to be solid. One clear result is that these satellite sequences change very rapidly from one species to another, and while this is hardly a new result, it is nevertheless very thoroughly documented here. The observation that genome size is not significantly correlated with satellite abundance seems at odds with what this author thought was generally accepted, but maybe that is only at a broader scale of species. The interpretation and significance of the findings is generally pretty thin, but there may not be much to say without further analysis.
The observation that DINEs are turned into satellite arrays further reinforces previous observations of the volatile nature of TE tandem arrays. We suggest only a few minor revisions and encourage the authors to proofread the manuscript and make edits for concision and typos.
Minor revisions:
Figure 3: It is unclear whether repeat length means monomer size or array length. Please use more clear language in reference to these analyses.
The author's should discuss whether satellites that are unique to one species are truly unique or if the sequences were in such low abundances in the genome that they were excluded by the filtering criteria that were applied.
In general the manuscript appears to be hastily prepared, and should have been proofread more carefully. There were several errors that might pass a spell checker, but not a careful reader, such as "Interesting, high sequence identity is limited to the first 30 bp" (line 283-284). They clearly meant, "Interestingly, …"?
Lines 308-311 got a bit jumbled.
Author Response
We thank the reviewer for all the suggestions. Below we provide a point-by-point response to all comments:
- Figure 3: It is unclear whether repeat length means monomer size or array length. Please use more clear language in reference to these analyses.
In the Line 38 we now made clear that repeat length means monomer size. We also added this information again in the legend of Figure 3.
- The author's should discuss whether satellites that are unique to one species are truly unique or if the sequences were in such low abundances in the genome that they were excluded by the filtering criteria that were applied.
We agree with the reviewer that the denomination of species-restricted satellite DNAs in our MS is misleading. Accordingly, we have reformulated the relevant part of the text as follows (lines 290-294): “In fact, our results showed that most satDNAs families (83%) seem to be restricted to a single species. However, our results obtained with TAREAN do not exclude the possibility that homologous low-copy number, or highly variable repeats are present in additional species.” We have deleted other parts of the text that mention species-restricted satellite DNAs, as well as figure 7.
-In general the manuscript appears to be hastily prepared, and should have been proofread more carefully. There were several errors that might pass a spell checker, but not a careful reader, such as "Interesting, high sequence identity is limited to the first 30 bp" (line 283-284). They clearly meant, "Interestingly, …"?
We apologize for that. The text has been extensively revised, so we hope to have eliminated most of the errors.
-Lines 308-311 got a bit jumbled.
We have rewritten this part of the text (lines 424-427) in the Conclusions section, also taking into account the suggestion made by referee 3.
Reviewer 3 Report
The manuscript touches an interesting and relevant topic within the field. Few minor corrections should be made:
-Line 70 please do not directly reference to [30] but rather use Conner et al. [30]
-same for the following line and other references within the manuscript. Please either address the first author or the group leader by name.
-The authors may consider shortening the Conclusion (e.g. omit in the present work, we aimed to ... ) in order to more briefly convey the conclusions of their findings.
Apart from this, the manuscript is well-written, the figures are concise.
Author Response
We thank the reviewer for all the suggestions. Below we provide a point-by-point response to all comments:
-Line 70 please do not directly reference to [30] but rather use Conner et al. [30]
We have reviewed the paper to correct all these issues concerning the references cited in the text.
-same for the following line and other references within the manuscript. Please either address the first author or the group leader by name.
We have corrected this issue.
-The authors may consider shortening the Conclusion (e.g. omit in the present work, we aimed to ... ) in order to more briefly convey the conclusions of their findings.
We shortened the Conclusion according to the referee’s suggestion (lines 424-430).